# Physiological specialization of the brain in bumble bee castes: Roles of dopamine in mating-related behaviors in female bumble bees

**Ayaka Morigami, Ken Sasaki** *

Graduate School of Agriculture, Tamagawa University, Machida, Tokyo, Japan

* sasakik@agr.tamagawa.ac.jp

**Data Availability Statement:** All relevant data are within the manuscript and its Supporting information files.

## Abstract

We aimed to investigate the roles of dopamine in regulating caste-specific behaviors in bumble bees and mating-related behaviors in bumble bee gynes. We examined caste differences in behaviors, biogenic amine levels, and expression levels of genes encoding dopamine receptors in the brains of bumble bees, and analyzed the effects of dopamine-related drugs on bumble bee behavior. Locomotor and flight activities were significantly higher in 8-day-old gynes and light avoidance was significantly lower in 4–8-day-old gynes than in same-aged workers. Brain levels of dopamine and octopamine were significantly higher in 8-day-old gynes than in same-aged workers, but tyramine and serotonin levels did not differ between the castes. Relative expression levels of the dopamine receptor gene *Big-Dop1* were significantly lower in 8-day-old gynes than in same-aged workers, but expression levels of other dopamine receptor genes did not differ between castes. Dopamine significantly enhanced locomotor and flight activities in 7–9-day-old workers, whereas the dopamine receptor antagonist flupentixol inhibited flight activity and mating acceptance in same-aged gynes. These results suggest that dopamine plays important roles in gyne-specific behavior in bumble bees and has a common dopaminergic function in female eusocial bees.

## Introduction

Phenotypic plasticity refers to the ability of a single genotype to produce different phenotypes when exposed to different environments [1, 2]. This manifests as the production of task-specialized individuals in social insects. Reproductive division of labor among nest members (castes) in social insects involves phenotypic plasticity, resulting in different patterns of behavior and physiological characteristics, which contribute to increasing the size of the nest and producing more reproductive individuals [1–4]. The differences in behavior patterns between castes assume different neural activities based on the brain physiology and morphology of each caste. Clarification of the brain physiological mechanisms underlying caste-specific behavior is therefore required to understand caste-specific brain differentiation in social insects.

**Funding:** This work was supported by the Japan Society for the Promotion of Science (JSPS) KAKENHI grant numbers JP20K06077 (https://kaken.nii.ac.jp/grant/KAKENHI-PROJECT-20K06077/) to KS. The funders had no role in study design, data collection and analysis, decision to publish, or preparation of the manuscript.

Biogenic amines are neuroactive substances that modulate the threshold of neurons and synaptic transmission efficiency in the central nervous system [5–7]. These effects contribute to the different behavioral phenotypes between castes in social insects. In the honey bee *Apis mellifera*, dopamine levels in the brain are significantly higher in virgin queens than in same-aged workers [8]. Higher dopamine levels in the brains of virgin queens maintain queen-specific behavior [9, 10]. Dopaminergic signaling is also associated with sugar responsiveness [11], memory retrieval in an appetitive context [12], appetitive motivation [13, 14] and aversive learning [15, 16] in honey bee workers. These behavioral roles of dopaminergic signaling may contribute to caste-specific behavior in honey bees. However, although the relationship between caste-specific physiological states and behavior has been reported in honey bees [8–10], the role of dopamine in other species of social Hymenoptera still needs to be determined.

Bumble bees comprise a group of eusocial bees that are phylogenetically separate from honey bees [17]. *Bombus ignitus* is a temperate annual species of bumble bee mainly distributed in Eastern Asia [18]. Gynes of this species are usually produced from the mature colony after male emergence in the autumn, and then leave the nest to mate with males, hibernate during winter, and then found a new nest the following spring [19]. The behavior of gynes during nest-founding is similar to that of female solitary bees, except that once the colony is established, the queens engage in oviposition and do not forage or care for the brood. The brain physiology of newly emerged gynes differs from that of emerged workers, with significantly higher levels of dopamine-related substances in the brains of gynes compared with workers [20]. Expression levels of nutrition-related genes in the brain were also shown to be significantly higher in gynes than in workers [20]. However, the ways in which these differences in physiological characteristics between castes affect their behaviors during the initial adult stages remain unknown.

Regarding the physiological specialization of the brain in different bumble bee castes, we hypothesized that the higher levels of dopamine-related substances in the brains of gynes might contribute to the activation of gyne-specific behaviors, especially mating-related behaviors. Mating behavior in gynes is based on basic behavioral activities such as locomotion and flight, light preference, and mating acceptance. To test the hypothesis, we first compared these behavioral activities between castes during the initial adult stages, and then tested the behavioral effects of applying dopamine or a dopamine receptor antagonist. We also examined caste differences in levels of dopamine and other monoamines, and in expression levels of dopamine receptor genes in the brain. The results of this study suggest that dopamine plays a role in the activation of mating-related behavior in gynes of *B. ignitus* and that there is physiological specialization of the brain in bumble bee castes.

## Materials and methods

### Bumble bee females

Commercially reared bumble bee (*B. ignitus*) colonies were maintained in wooden boxes (length 30 cm × width 20 cm × height 12 cm) with transparent windows at the top and steel nets at the bottom at 28°C in constant darkness. Each wooden box was divided in two sections: one (length 18 cm × width 20 cm × height 12 cm) for the nest and the other (length 12 cm × width 20 cm × height 12 cm) for feeding with fructose and sucrose solution. Bees were given *ad libitum* pollen kneaded with sugar solution in the nest room and sugar solution in the feeding room. Newly emerged workers were taken from developing colonies (during the late social phase) that had produced 40–50 workers, but no males or virgin gynes. We selected similar middle-sized individuals among the newly emerged workers and excluded extremely small or large workers, to reduce variations in body size in our sample. Newly emerged gynes were

collected from a mature colony producing males. Typical workers and gynes were therefore collected from 15 colonies at different colony-growth stages. Both newly emerged workers and gynes were marked on the thorax and returned to the mother colonies, and then recollected for behavioral experiments. Some 6–8-day-old workers and gynes were collected for measurements of biogenic amines and expression levels of dopamine receptor genes. The heads of workers and gynes were stored in liquid nitrogen before measurements of biogenic amines or gene expression analyses.

## Behavioral activity in gynes and workers

Workers and gynes collected from the mother colonies were individually transferred to a ring-shaped stainless-steel chamber (outer diameter 150 mm; inner diameter 90 mm; height 30 mm) with a transparent sheet cover at 28˚C until the behavioral experiments (Fig 1A). The transparent cover sheet was separated into four divisions by crosshairs, and half of the transparent cover was then covered by a red plastic sheet to provide a dark (red) area in the chamber. After acclimatization for 5 min, the spontaneous locomotor activity of each bee was recorded for 15 min using a digital video camera. The number of times an individual crossed the crosshairs was counted to determine locomotion, and the time spent under the red plastic sheet was recorded to determine light avoidance (or preference).

Flight behavior was observed in a net cage (45 cm × 60 cm × 45 cm) at 28˚C (Fig 1B). The chamber containing a female bumble bee was fully covered by red plastic sheet and transferred into the net cage. The red sheet was then removed to allow the female to fly spontaneously. The flight behavior was observed for 5 min after removing the red sheet and classified according to three levels: level 1 (not flown and not flapped wings), level 2 (not flown but flapped wings), and level 3 (flown).

## Measurements of biogenic amines

Bumble bee heads stored in liquid nitrogen were dissected in 0.1 M phosphate buffer (pH 7.0) on a Peltier cooling unit (Kenis Ltd., Osaka, Japan) at approximately 4˚C under a microscope.

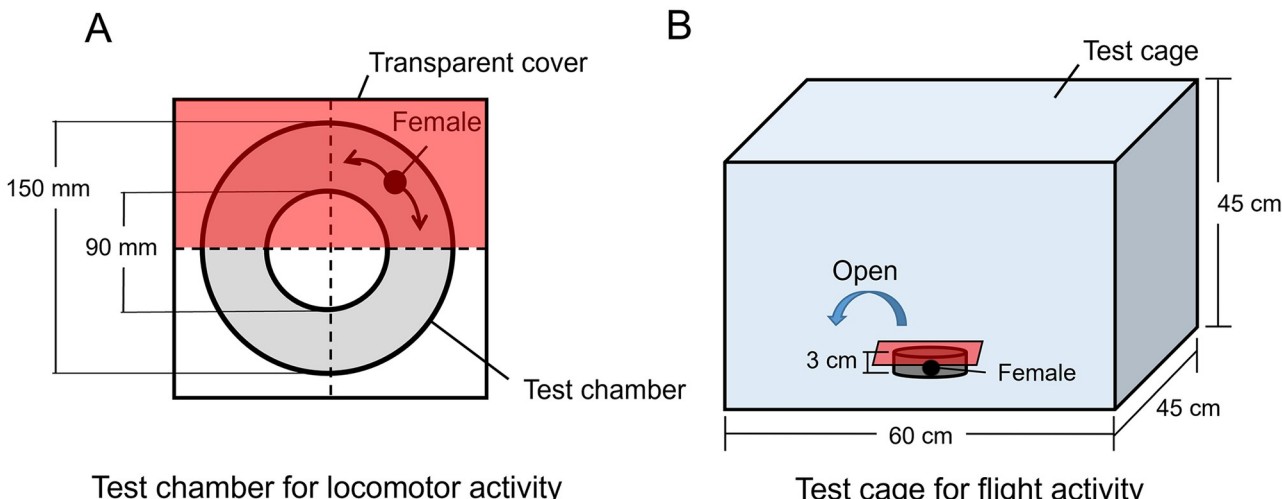

**Fig 1. Equipment for measuring locomotor and flight activities.** (A) Stainless-steel test chamber for locomotor activity. A female *Bombus ignitus* can walk freely within the annular circuit (gray area), which was covered by a transparent cover sheet. The sheet was separated into four sections by crosshairs (dotted lines), and half of the transparent cover was covered by a red plastic sheet to provide a dark (red) area in the chamber. The number of times the crosshairs were crossed was counted. (B) Test cage for flight activity. The red plastic sheet was removed from the test chamber and the duration of flight initiation was recorded.

Each dissected brain was homogenized for 2 min with a microglass homogenizer in 100 μL of ice-cold 0.1 M perchloric acid containing 0.1 ng/μL 3, 4-dihydroxybenzylamine. Each single brain sample was then transferred into a 1.5 mL microcentrifuge tube and centrifuged at $15,000 \times g$ at 4°C for 30 min. Each supernatant was transferred into microvials for high-performance liquid chromatography-electrochemical detection (HPLC-ECD) of dopamine, serotonin, tyramine, and octopamine levels. The HPLC-ECD system was developed by Sasaki et al. [8] and comprised a solvent delivery pump (PU-4180, JASCO, Tokyo, Japan), refrigerated automatic injector (AS-4050, JASCO), and C18 reversed-phase column (250 mm × 4.6 mm internal diameter, 5 μm average particle size; UG120, Osaka Soda, Osaka, Japan), and the temperature was maintained at 35°C. The ECD (ECD-700, EICOM, Kyoto, Japan) was set at 0.85 V and employed at 35°C. The mobile phase contained 0.18 M monochloroacetic acid and 40 μM 2Na-EDTA, adjusted to pH 3.6 with NaOH, and 1.62 mM sodium-1-octanesulfonate and 5% $CH_3CN$ were added to this solution. The flow rate was kept constant at 0.7 mL/min.

External standards were run in the HPLC-ECD system before and after the sample runs to identify and quantify biogenic amines. The peaks were identified by comparing the retention time and hydrodynamic voltammograms with those obtained for the standards. Measurements based on the peak areas in the chromatograms were obtained by calculating the ratio of the peak area for a substance relative to the peak area for the standard.

Protein levels in each brain were measured using the Bradford method [21] and levels of biogenic amines were normalized based on the brain-protein contents. The precipitated protein pellet obtained from the brain tissue after extracting biogenic amines was treated as described by Sasaki et al. [8]. The absorbance of each reacted sample was measured at a wavelength of 590 nm using a microplate reader (MPR-A100, AS ONE, Osaka, Japan).

## Measurements of relative expression levels of dopamine receptor genes

Frozen brains were dissected in 0.1 M phosphate buffer on a cooling unit at approximately 4°C under a dissecting microscope. Total RNA was extracted from two pooled brains using ISOGEN (NipponGene, Tokyo, Japan). The dissected brains were then homogenized using an electric homogenizer (T10+S10N-5G; IKA Works, Staufen, Germany) according to the manufacturer's instructions. During the extraction procedure, the RNA was treated with rDNase (RT Grade for Heat Stop, NipponGene) for 15 min to remove genomic DNA, and then mixed with stop solution at 70°C for 10 min. The quality and quantity of the extracted RNA were determined at 230, 260, and 280 nm using a microvolume spectrophotometer (Nano-Drop2000; Thermo Fisher Scientific, Waltham, MA, USA). Single-stranded cDNA was synthesized by reverse transcription of DNase-treated RNA (500 ng) using a high-capacity cDNA Reverse Transcription kit (Applied Biosystems, Waltham, MA, USA) according to the manufacturer's instructions. Negative control samples without reverse transcriptase were treated using the same procedure.

Four genes encoding dopamine receptors (*BigDop1*, *BigDop2*, *BigDop3*, and *BigDopEcR*) were selected as the targets for real-time quantitative polymerase chain reaction (RT-qPCR) (S1 Table) [20]. Actin 5C (*BigACT*), glyceraldehyde-3-phosphate dehydrogenase 2 (*BigGAPDH*), 40S ribosome protein S3 (*BigRPS3*), and ribosome protein 49 (*BigRP49*) were tested as reference genes using the primers shown in S1 Table [20, 22]. Standard curves were generated for each target and reference gene with a series of template DNA dilutions (1, 1/5, 1/10, and 1/20 times). The standard curves were based on the relative concentration of cDNA and the quantification cycle (Cq) required for each RT-qPCR to cross a threshold fluorescence intensity within the linear portion of the amplification curve. RT-qPCR was performed using a KAPA SYBR FAST qPCR kit (KAPA Biosystems, Nippon Genetics, Tokyo, Japan) with a RT-

qPCR system (Eco, Illumina, San Diego, CA, USA). Each reaction mixture (total volume of 20 μL) comprised 10 μL of KAPA SYBR Universal qPCR mix, 0.4 μL each of the forward and reverse primers (10 μM), 7.2 μL RNase-free water, and 2 μL of the cDNA template. The temperature profile for amplifying the target gene and reference gene fragments was 95˚C for 1 min, followed by 40 cycles at 95˚C for 3 s and 60˚C for 2 s. Each individual sample was repeated three times in a single RT-qPCR run. Amplification of the single product was confirmed by dissociation-curve analysis using the RT-qPCR system.

We recorded the Cq values for the reference and target genes to estimate the mRNA expression levels of the genes encoding dopamine receptors. The stabilities of the internal control reference genes (*BigACT*, *BigGAPDH*, *BigRPS3*, *BigRP49*) were evaluated using Best Keeper software [23] and analyzed by the Mann-Whitney U-test. The Cq values for *BigACT* were most stable in Best Keeper, with no significant differences between the groups (Mann-Whitney U-test, $z = -1.209$, $P = 0.227$, 10 samples each). We therefore normalized the expression levels of the target genes using the expression levels of *BigACT*. The analyses were performed according to the Minimum Information for Publication of Quantitative Real-Time PCR Experiments (MIQE) guidelines [24].

## Effects of dopamine and dopamine receptor antagonist on behavior

We examined the roles of dopamine in the locomotion, light avoidance, and flight behaviors of bumble bee females by applying dopamine or a dopamine receptor antagonist. The experiment was designed so that females with lower behavioral activity were injected with dopamine, and females with higher behavioral activity were injected with a dopamine receptor antagonist. Based on comparisons of behavioral activity between castes (see Results), workers were injected with dopamine and gynes with a dopamine receptor antagonist. Workers (7–9-days-old) were injected into the abdomen with 2 μL of dopamine solution (Sigma-Aldrich, St. Louis, MO, USA) ($10^{-3}$ M and $10^{-2}$ M) dissolved in 0.1 M phosphate buffer (pH 7.0) using a 10-μL microsyringe with a fine needle. Control workers were injected with 2 μl of 0.1 M phosphate buffer. Gynes (7–9-days-old) were injected into the abdomen with 2 μL of flupentixol solution (dopamine receptor antagonist; Sigma-Aldrich) ($10^{-3}$ M and $10^{-2}$ M) dissolved in 0.1 M phosphate buffer, and control gynes were injected with 2 μl of 0.1 M phosphate buffer. Dopamine injections were carried out with and without cooled anesthesia on ice for 20 min. Bees were acclimatized for 40 min after dopamine injection before observing locomotor activities.

The role of dopamine in mating acceptance in gynes was determined by injecting 7–9-day-old virgin gynes into the abdomen with 2 μL of flupentixol solution, using the same procedures as for the behavioral experiments described above. The gynes were acclimatized for 5 min after drug injection and then transferred to a net cage (45 cm × 60 cm × 45 cm) at 28˚C to allow them to mate voluntary with males. Twice as many males as gynes were prepared and transferred into the cage for the mating experiment, and the number of gynes that mated with males was counted for 60 min.

## Statistical analysis

The data did not meet the criteria for parametric tests and nonparametric tests were therefore used for these analyses. Locomotor activity and light avoidance were compared between individuals of the same age in different castes using the Mann-Whitney U-test (significant value $P = 0.05$). The proportions of flying individuals were compared between castes of the same age using Fisher's exact test (significant value $P = 0.05$).

Brain levels of biogenic amines and relative expression levels of dopamine receptor genes were compared between castes using Mann-Whitney U-tests.

Locomotor activity and light avoidance were compared between bees injected with different concentrations of dopamine or flupentixol (control, $10^{-3}$ M, and $10^{-2}$ M) using the Kruskal-Wallis test with the Steel test (control vs. $10^{-3}$ M, control vs. $10^{-2}$ M, significant value $P = 0.05$). These data were also analyzed by Spearman's rank correlation test. The proportions of flying and mated individuals were compared among bees injected with different concentrations of dopamine or flupentixol using Fisher's exact test and logistic regression analysis. For logistic regression analysis, a model of the logistic function between concentrations of drug and behavioral reactions (flying vs. not flying, mated vs. not mated) was assumed.

## Results

### Caste differences in behavioral activities

Workers and gynes showed similar levels of locomotor activity at 0 and 2 days after emergence, indicated by numbers of crossings in the arena (Mann-Whitney U-test, 0 day: $z = -1.109$, $P = 0.267$; 2 days: $z = -0.485$, $P = 0.627$), but locomotor activity was significantly higher in gynes than in workers at 4, 6, and 8 days (4 days: $z = -2.333$, $P < 0.05$; 6 days: $z = -2.494$, $P < 0.05$; 8 days: $z = -2.938$, $P < 0.01$; Fig 2A, S2 Table).

Light avoidance, indicated by the duration of stay in the red area, was also similar between castes at 0 and 2 days after emergence (Mann-Whitney U-test, 0 day: $z = -0.699$, $P = 0.485$; 2 days: $z = -0.471$, $P = 0.638$), but was significantly lower in gynes than in workers at 4, 6, and

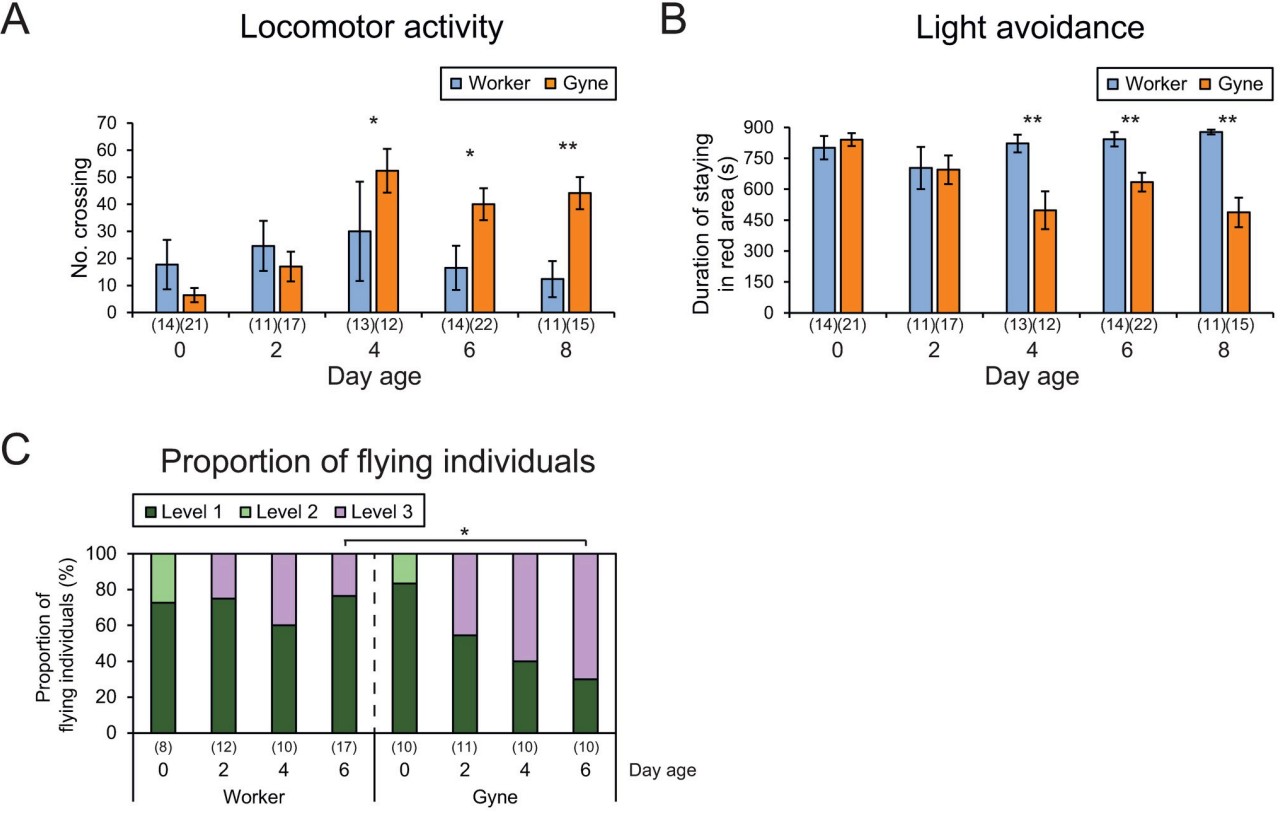

**Fig 2. Caste differences in behavioral activities in bumble bees.** (A) Locomotor activity, (B) light avoidance, and (C) proportion of flying individuals. Numbers in parentheses indicate numbers of samples. Significant differences between castes at the same age were examined by Mann-Whitney U-test (*$P < 0.05$, **$P < 0.01$).

8 days (4 days: $z = -3.119$, $P < 0.01$; 6 days: $z = -3.311$, $P < 0.01$; 8 days: $z = -3.842$, $P < 0.001$; Fig 2B, S2 Table).

The proportion of flying individuals (level 3) did not differ significantly between workers and gynes at 0, 2, or 4 days old (Fisher's exact test, 3 × 2 table, 0 day: $\chi^2 = 0.379$, $P = 0.64$, 2 days: $\chi^2 = 1.059$, $P = 0.4$, 4 days: $\chi^2 = 0.8$, $P = 0.656$), but the proportion was significantly higher in gynes than in workers at 6 days old (Fisher's exact test, $\chi^2 = 5.632$, $P < 0.05$; Fig 2C, S2 Table).

Locomotor and flight activities and light preference were thus similar between castes at the early adult stage, but became higher in gynes than in workers after 6 days old.

### Biogenic amine levels in different castes

Dopamine levels in the brain were significantly higher in 8-day-old gynes than in same-aged workers (Mann-Whitney U-test, $z = -3.065$, $P < 0.01$; Fig 3A, S3 Table). Brain levels of serotonin and tyramine did not differ between 8-day-old individuals of different castes ($z = -1.385$, $P = 0.166$; $z = -1.771$, $P = 0.077$, respectively; Fig 3B and 3C, S3 Table), but levels of the

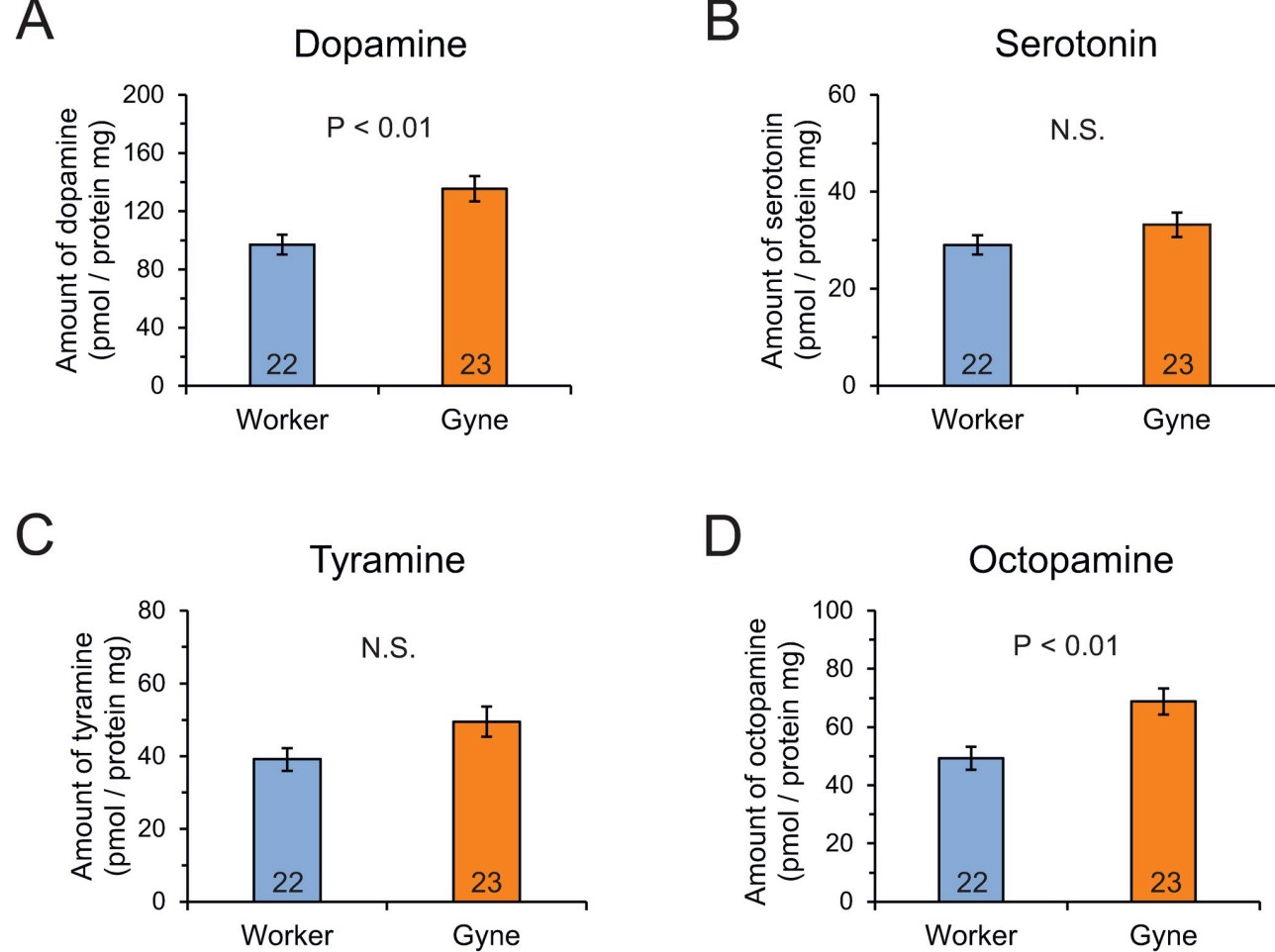

**Fig 3. Biogenic amine levels in the brain in workers and gynes in bumble bees.** (A) Dopamine, (B) serotonin, (C) tyramine, and (D) octopamine. Numbers in bars indicate numbers of samples examined. Significant differences between castes were examined by Mann-Whitney U-test.

tyramine metabolite octopamine were significantly higher in gynes than in workers at 8 days after emergence ($z = -3.088$, $P < 0.01$; Fig 3D, S3 Table).

### Expression levels of dopamine receptor genes

Relative expression levels of dopamine receptor genes (*BigDop1*, *BigDop2*, *BigDop3* and *BigDopEcR*) in the brain were determined by RT-qPCR (S4 Table). Expression levels of *BigDop1* were significantly lower in gynes than in workers (Mann-Whitney U-test, $z = -1.965$, $P < 0.05$; Fig 4A), but expression levels of the other three tested genes did not differ significantly between castes (*BigDop2*: $z = -1.890$, $P = 0.059$; *BigDop3*: $z = -1.587$, $P = 1.112$; *BigDopEcR*: $z = -0.302$, $P = 0.762$; Fig 4B–4D), although levels of *BigDop2* were almost significant ($P = 0.059$).

### Effects of dopamine injections on behavioral activity

Locomotor activity did not differ between individuals injected with different concentrations of dopamine (control, $10^{-3}$ M and $10^{-2}$ M) in the "not cooled" (Kruskal-Wallis test, $H = 4.497$, $P = 0.106$) and "cooled" treatment groups ($H = 1.968$, $P = 0.374$), but locomotor activity overall

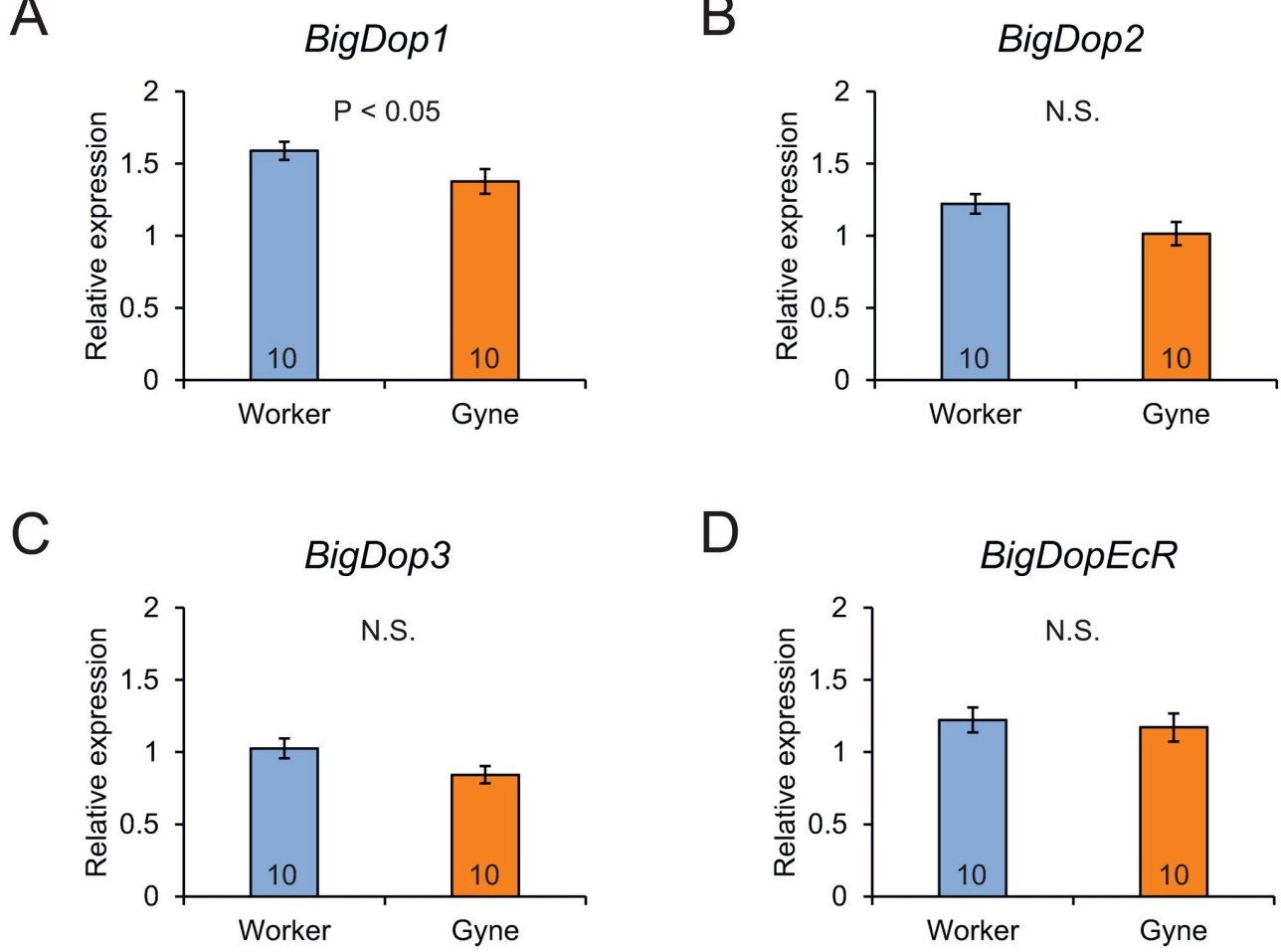

**Fig 4. Relative expression levels of genes encoding dopamine receptors.** (A) *Big Dop1*, (B) *BigDop2*, (C) *BigDop3*, and (D) *BigDopEcR*. Numbers in bars indicate numbers of samples examined. Significant differences between castes were examined by Mann-Whitney U-test.

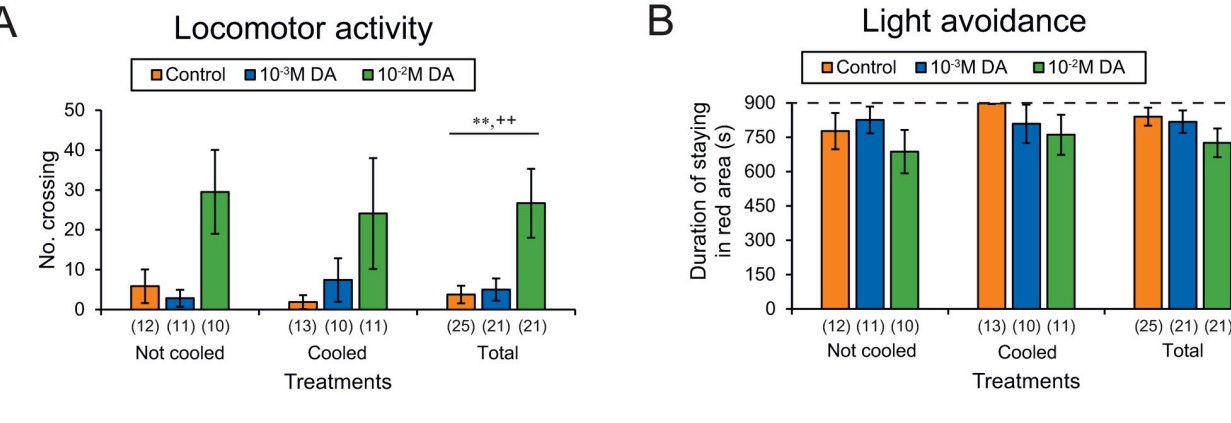

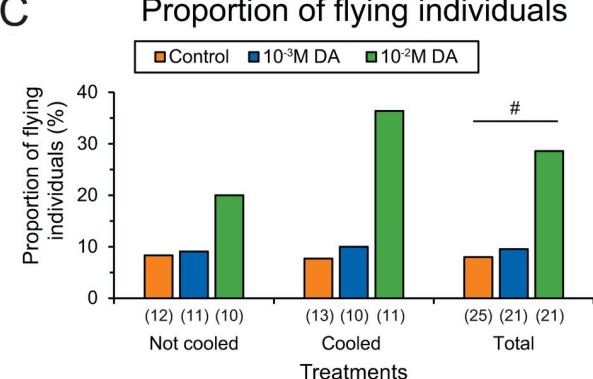

**Fig 5. Effects of dopamine injection into 7–9-day-old workers on behavioral activities.** (A) Locomotor activity, (B) light avoidance, and (C) proportion of flying individuals. Numbers in parentheses indicate numbers of samples examined. Differences among groups were analyzed by the Kruskal-Wallis test (**$P < 0.01$). Significant correlations with dopamine concentrations were tested by Spearman's rank correlation test (++$P < 0.01$) or logistic regression test (#$P < 0.05$). DA: dopamine.

was significantly higher in workers injected with $10^{-2}$ M dopamine compared with controls (Kruskal-Wallis test, $H = 12.056$, $P < 0.01$; Steel test, control vs. $10^{-2}$ M dopamine, $P < 0.05$; Fig 5A, S5 Table). There was a significant positive correlation between the number of crossings and the injected dopamine concentration in workers overall (Spearman's rank correlation, $r_s = 0.402$, $P < 0.01$, $n = 67$).

Light avoidance behavior did not differ between individuals injected with different concentrations of dopamine in the "not cooled" (Kruskal-Wallis test, $H = 2.541$, $P = 0.281$) and "cooled" treatment groups ($H = 1.963$, $P = 0.375$) and overall ($H = 3.599$, $P = 0.165$) (Fig 5B, S5 Table). There was no significant correlation between the duration of stay in the red area and injected dopamine concentrations in workers overall (Spearman's rank correlation, $r_s = -0.214$, $P = 0.083$, $n = 67$).

The proportion of flying individuals (level 3) did not differ significantly between individuals injected with different concentrations of dopamine in the "not cooled" (Fisher's exact test, $2 \times 3$ table, $\chi^2 = 0.839$, $P = 0.661$) and "cooled" treatment groups ($\chi^2 = 3.94$, $P = 0.139$) and overall ($\chi^2 = 4.507$, $P = 0.157$; Fig 5C, S5 Table). However, logistic regression analysis of "flying" and "not flying" individuals indicated that the flying individuals significantly increased with increasing concentration of dopamine injected ($r^2 = 0.074$, $\chi^2 = 4.077$, $P < 0.05$, $n = 67$).

### Effects of dopamine receptor antagonist on behavioral activity

Locomotor activity did not differ significantly in relation to the concentration of dopamine receptor antagonist flupentixol (control, $10^{-3}$ M and $10^{-2}$ M) in the "not cooled" (Kruskal-Wallis test, $H = 4.336$, $P = 0.114$) and "cooled" treatment groups ($H = 2.377$, $P = 0.305$) and overall ($H = 2.707$, $P = 0.258$; Fig 6A, S6 Table). There was no significant correlation between the number of crossings and flupentixol concentration in gynes overall (Spearman's rank correlation, $r_s = -0.192$, $P = 0.108$, n = 71).

Light avoidance behavior was also similar in individuals treated with different concentrations of flupentixol in the "not cooled" (Kruskal-Wallis test, $H = 4.336$, $P = 0.114$) and "cooled" treatment groups ($H = 0.889$, $P = 0.641$) and overall ($H = 0.458$, $P = 0.795$; Fig 6B, S6 Table). There was no significant correlation between the duration of stay in the red area and flupentixol concentration in gynes overall (Spearman's rank correlation, $r_s = 0.08$, $P = 0.503$, $n = 71$).

The proportion of flying individuals (level 3) did not differ significantly in relation to the injected flupentixol concentration in the "not cooled" (Fisher's exact test, 2 × 3 table, $\chi^2 = 3.685$, $P = 0.201$) and "cooled" treatment groups ($\chi^2 = 2.482$, $P = 0.318$) and overall ($\chi^2 = 4.466$, $P = 0.104$; Fig 6C, S6 Table). However, logistic regression analysis indicated that flying individuals decreased significantly in line with decreasing concentrations of flupentixol ($r^2 = 0.062$, $\chi^2 = 3.846$, $P < 0.05$, $n = 71$).

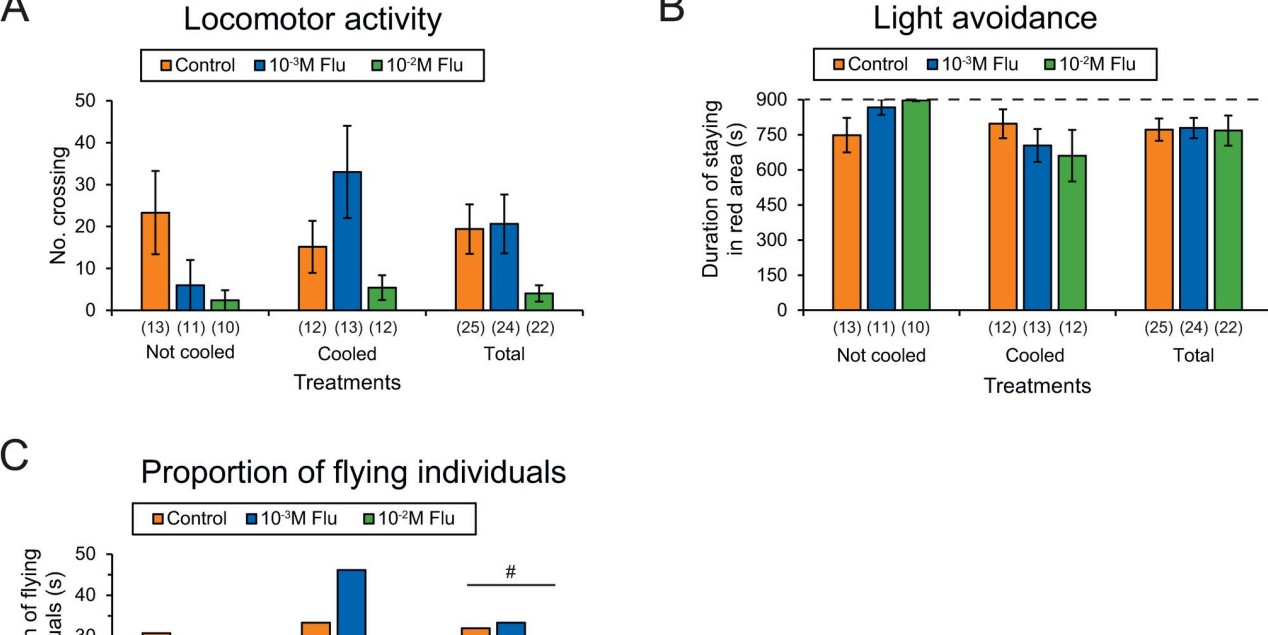

**Fig 6. Effects of dopamine receptor antagonist flupentixol injection into 7–9-day-old gynes on behavioral activities.** (A) Locomotor activity, (B) light avoidance, and (C) proportion of flying individuals. Numbers in parentheses indicate numbers of samples examined. Significant correlations with flupentixol concentrations were tested by logistic regression test (#$P < 0.05$). Flu: flupentixol.

# Proportion of mated individuals

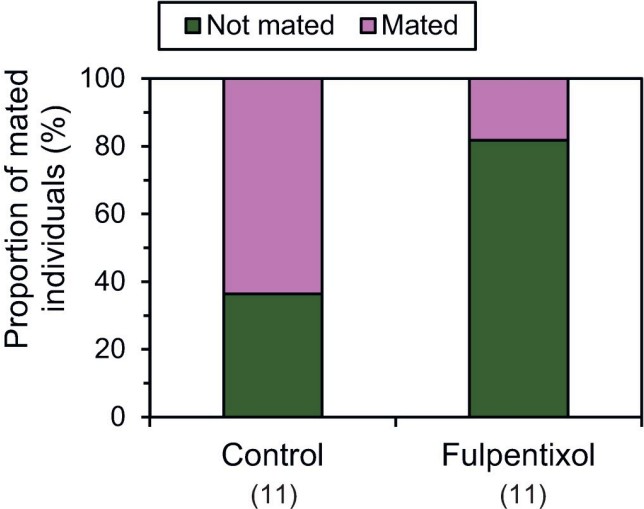

**Fig 7. Effects of dopamine receptor antagonist flupentixol injection into 7–9-day-old gynes on mating acceptance.** Numbers in parentheses indicate numbers of samples examined.

## Effects of dopamine receptor antagonist on mating acceptance

Gynes accepted mating by a male or rejected it by bending their abdomen ventrally. Injection of flupentixol tended to decrease the proportion of mated individuals. The proportion was not different according to Fisher's exact test ($\chi^2 = 4.7$, $P = 0.08$; Fig 7, S7 Table), but logistic regression analysis detected a difference between the control and flupentixol groups ($r^2 = 0.165$, $\chi^2 = 4.242$, $P < 0.05$, $n = 22$).

## Discussion

We tested the hypothesis that dopamine plays an important role in promoting mating behavior in females and contributes to gyne-specific behavioral states in bumble bees. We accordingly investigated caste differences in behavioral activities and brain levels of monoamines and dopamine receptor gene expression, and examined the behavioral effects of dopamine and dopamine receptor antagonist treatment. The results showed that behavioral activities and dopamine levels differed between castes, while dopamine promoted locomotor and flight activities, and the dopamine antagonist flupentixol inhibited flight activity and mating acceptance. These results supported a role for dopamine in gyne-specific behaviors, and suggested the importance of caste-specific dopamine levels in the brain in bumble bees.

### Caste-specific behavioral activities and dopaminergic system

Bumble bee workers and gynes appear at different stages of colony development; workers are produced from colony founding to the production of reproductive individuals, whereas gynes emerge after male emergence. Gynes in the nest do not show typical worker behaviors, including nursing larvae, storing honey, and foraging, but remain in the nest until they leave to mate with males from other nests. The behaviors of gynes before mating thus differ from those of workers; however, these behaviors have not been described quantitatively and have not been examined in behavioral tests. The current results showed that locomotor and flight activities

gradually increased in gynes and were significantly higher in 4-, 6-, and 8-day-old gynes compared with same-aged workers. In addition, light preference was significantly higher in gynes at these ages compared with same-aged workers. These results indicated that gynes required higher activity levels and light preference to allow them to leave the nest and mate with males. These behavioral parameters can thus be used to determine the effects of dopamine-related drug applications.

Levels of the above behavioral activities were high in 8-day-old gynes and corresponded to the higher brain levels of dopamine and octopamine. Higher levels of dopamine and tyramine have previously been found in the brains of newly emerged gynes [20], suggesting that higher dopamine levels in gynes may be maintained from emergence to 8 days old, whereas the high tyramine levels at emergence might shift to higher octopamine levels in 8-day-old gynes as a result of the conversion of tyramine to octopamine. The current results demonstrated the contributions of dopamine to gyne behavior; however, the effects of octopamine remain unknown.

We also explored caste differences in dopaminergic signaling by determining the relative expression levels of genes encoding dopamine receptors. Whole-brain expression levels of *Big-Dop1* were significantly lower in gynes than in workers, while levels of *BigDop2* also tended to be lower in gynes, but the difference was not significant ($P = 0.059$). These results seemed inconsistent with the active dopamine signaling state in gynes; however, it is possible that the brain regions expressing dopamine receptor genes are more restricted in gynes than in workers. Similar lower expression levels of *Dop1* and *Dop2* have been reported in queens and reproductive workers in honey bees [8, 25], and lower levels of *Dop1* were found in reproductive females in the ant *Harpegnathos saltator* [26]. Further detailed analyses of the brain regions expressing these genes are therefore required.

This study determined biogenic amine levels and expression levels of dopamine receptor genes in the whole brain and examined the effects of dopamine-related drug injections. In terms of behaviors, the whole brain acts as a functional unit in insects, integrating multimodal sensory inputs detected by several sense organs and generating motor outputs to drive several effector organs. This study thus adopted a whole-brain approach. Detailed analyses based on specific brain regions may be more informative than the whole-brain approach; however, degradation of biogenic amines or mRNA and loss of these substances from the cut surface of the brain during dissection could occur in a brain-region-based approach. We therefore considered that a whole-brain approach was a suitable first step that should be followed by a brain-region-based approach. Biogenic amine levels in specific brain regions have previously been reported in honey bees [27, 28], and gene expression has also been reported using a brain-region-based approach in honey bees [29–31] and bumble bees [32]. Dopamine levels and expression levels of dopamine receptor genes in specific brain regions should thus be compared between castes in *B. ignitus* in future studies.

## Roles of dopamine in behavior

Treatment of workers with dopamine enhanced their locomotor and flight activities but had no significant effect on their light preference, while the treatment of gynes with a dopamine receptor antagonist inhibited their flight and mating acceptance behaviors. These results suggested that dopamine could modulate locomotor and flight activities in females, which might also influence mating acceptance. Dopamine levels in the brain were significantly higher in gynes than in workers at emergence [20] and at 8 days old (present study). These higher dopamine levels may enhance the locomotor and flight activities and influence mating acceptance during the initial adult stage. It is essential for gynes to complete mating with males during the

reproductive period because they may not be able to meet the males after hibernation, resulting in the production of only unfertilized (male) eggs during nest founding. Virgin gynes may thus possess a dopamine-based mechanism to promote basic behavioral activities during the initial adult stage. After mating, dopamine levels in gynes gradually decrease to the nest-founding stage via low temperature diapause [33]. These temporal changes in dopamine levels in virgin gynes may thus function to ensure reliable mating.

The relatively higher levels of dopamine in the brain of young virgin gynes may contribute to the activation of mating-related behaviors. However, dopamine may also affect behavioral activation in workers, especially foragers that fly out of the nest, if foragers have relatively higher dopamine levels than other workers. Dopamine may potentially act in all females, but its effects may be controlled by its relative levels in the brain. Dopamine levels in foragers and other workers remain to be determined.

Dopamine plays other roles in behavior of social insects [10, 34, 35]. In honey bees, it mediates a decrease of sucrose responsiveness [11; but see 13 in Fig 3], affects memory retrieval in an appetitive context [12, 13], regulates appetitive motivation [13, 14] and mediates aversive reinforcement signaling in aversive learning [15, 16]. In ants, dopamine is relevant for the defense of the territory or colony [35]. These roles of dopamine may contribute to caste-specific behavior in *B. ignitus*. Behavioral differences between bumble bee castes should thus be investigated in more detail in relation to dopamine application.

Caste differences in brain dopamine levels in the initial adult stage have been reported in honey bees [8], bumble bees [20], and ants [26, 36]. In these species, higher levels of dopamine in the brain are found in reproductive females and contribute to the promotion of mating-related behavior [10] or aggressive behaviors against rival virgin queens [9] or potential rival females [37]. Caste-specific dopamine levels in the brain thus function to maintain caste-specific behaviors in several species of Hymenoptera.

## Conclusion

We detected caste differences in behaviors at the initial adult stage and determined the roles of dopamine in mating-related behaviors in bumble bees. Gynes had higher locomotor and flight activities and lower light avoidance than workers during the initial adult stages. Gynes also had higher brain levels of dopamine and octopamine at 8 days old, but lower relative expression levels of the dopamine receptor gene *BigDop1* compared with same-aged workers. Dopamine treatment enhanced locomotor and flight activities in workers, whereas treatment with a dopamine receptor antagonist inhibited flight activity and mate acceptance in gynes. These results suggested that higher dopamine levels in gynes were associated with increased locomotion, flight, and mating activities. Caste differences in brain physiology during the initial adult stages thus led to caste-specific behaviors in bumble bees.

## Supporting information

**S1 Table. Primer sequence of the genes for RT-qPCR.**
(PDF)

**S2 Table. Data of behavioral activities in workers and gynes.**
(PDF)

**S3 Table. Biogenic amine levels in the brain in workers and gynes.**
(PDF)

**S4 Table. Relative expression levels of genes encoding dopamine receptors in the brain in workers and gynes.**
(PDF)

**S5 Table. Data of behavioral activities in workers with dopamine injections.**
(PDF)

**S6 Table. Data of behavioral activities in gynes with flupentixol injections.**
(PDF)

**S7 Table. Number of mated and unmated individuals.**
(PDF)

## Author Contributions

**Conceptualization:** Ken Sasaki.

**Data curation:** Ayaka Morigami.

**Formal analysis:** Ayaka Morigami, Ken Sasaki.

**Funding acquisition:** Ken Sasaki.

**Investigation:** Ayaka Morigami.

**Resources:** Ken Sasaki.

**Supervision:** Ken Sasaki.

**Writing – original draft:** Ken Sasaki.

**Writing – review & editing:** Ayaka Morigami.

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
