## [Decision Letter · Decision Letter 0]

14 Nov 2023

PONE-D-23-31847Physiological specialization of the brain in bumble bee castes: Roles of dopamine in mating-related behaviors in female bumble beesPLOS ONE

Dear Dr. Sasaki,

Thank you for submitting your manuscript to PLOS ONE. After careful consideration, we feel that it has merit but does not fully meet PLOS ONE’s publication criteria as it currently stands. Therefore, we invite you to submit a revised version of the manuscript that addresses the points raised during the review process.

In the reviews below you will see that both expert reviewers saw potential for the manuscript to be suitable for PLOS ONE, but also suggest some important revisions that are required to meet publication criteria. I have allocated Major revision to enable sufficient time.

We look forward to receiving your revised manuscript.

Kind regards,

Adrian G Dyer, Ph.D.

Academic Editor

PLOS ONE

Reviewers' comments:

Reviewer's Responses to Questions

**Comments to the Author**

1. Is the manuscript technically sound, and do the data support the conclusions?

Reviewer #1: Partly

Reviewer #2: Yes

2. Has the statistical analysis been performed appropriately and rigorously? 

Reviewer #1: Yes

Reviewer #2: Yes

3. Have the authors made all data underlying the findings in their manuscript fully available?

Reviewer #1: No

Reviewer #2: Yes

4. Is the manuscript presented in an intelligible fashion and written in standard English?

Reviewer #1: Yes

Reviewer #2: Yes

5. Review Comments to the Author

Reviewer #1: This manuscript studies the role of dopaminergic signalling in the regulation of caste specific behaviors in workers and gynes of bumble bees Bombus ignitus. The authors quantified biogenic amine levels (including dopamine) and expression levels of dopamine receptor genes (BigDop1 to 3 and BigDopEcr) in entire bee brains and related differences in these parameters to behavioral differences (flight and motor activity, light avoidance, mating acceptance). The main findings reported indicate that dopamine levels are higher in 8-day old gynes compared to 8-day workers and that this difference correlates with a higher flight and motor activity as well as with decreased light avoidance in gynes compared to workers. In parallel, a decrease in brain expression of BigDop1 was observed in gynes but not in workers. No differences were found for the other genes. Finally, a pharmacological approach was used to enhance dopamine levels in workers and to reduce them in gynes via the use of the dopamine receptor antagonist flupentixol. Enhancing dopamine levels increased motor and flight activity of 7-9 day old workers; reducing them in gynes of the same age reduced flight activity and mating acceptance.

The manuscript reports interesting data but has at the same time some methodological issues, which raise doubts about the soundness of some of its conclusions. These issues are detailed below.

1) Measurement of biogenic amines

First of all, it is unclear whether the measurements were performed on a single brain basis, and thus the values reported are the mean values of several individual brains, or whether they were obtained in homogenates of multiple (how many?) brains. This is not specified. Second, reporting HPLC values for entire brains may be misleading as an increase in a specific brain region may go hand-by-hand with a decrease in another brain region with a different role in mediating the same behavior. The result, in this case, would be a ‘no-change’ situation as one would compensate the other and thus, the entire dynamic variation would be lost. While important, massive changes in biogenic amine level could be indeed detected using the approach chosen, fine variations on brain region basis, which could be more informative, are definitely overlooked. The authors should be aware of this situation and discuss it properly. For a different, brain-region based approach see, for instance, [1, 2].

2) Measurement of gene expression levels

The same criticism applies to the approach used to quantify gene expression levels. In this case homogenates of two brains were used but no effort was made to section different brain regions and quantify gene expression levels on brain-region basis. This is particularly worrying as gene expression is known to change in different ways in different brain regions, according to their role and implication in several behaviors. It is not surprising, therefore, that no changes were detected for the majority of genes analyzed. The approach chosen may in fact be hiding significant changes in different regions according to behavioral state and caste, which are overlooked by this procedure. This analysis should be reconsidered.

For a different, brain-region based approach see, for instance, [3-6].

3) Pharmacological manipulation of dopamine levels

The authors chose to increase dopamine levels in workers via abdominal injection of dopamine, and to decrease dopamine levels in gynes via abdominal injection of flupentixol, an antagonist of dopamine receptors.

It is difficult to understand why this partial approach was selected instead of having used the double approach (increase and decrease) both in workers and in gynes. The authors should have injected dopamine and flupentixol in individuals of each caste. In this way, testing for necessity and sufficiency of dopaminergic signalling in the behaviors observed would be possible. Currently, the manuscript seems to provide an incomplete and partial analysis in each caste even if some interesting differences were found. It would be desirable to complete the missing approach in each caste (flupentixol in workers and dopamine in gynes) to provide a complete and accurate picture of the role of dopamine in caste-related behavior. One could for instance show that flight, motor activity and mating acceptance is enhanced in a precocious way in younger gynes.

Minor remarks

Introduction

The authors discuss the role of biogenic amines, and more specifically of dopamine, in the regulation of bee behavior (lines 30 -40). Yet, in doing this they reduce this discussion to their own work (6 from 9 references), omitting thereby significant findings on the role of dopamine in bees. This needs to be corrected to achieve a fairer introduction. The role of dopaminergic signalling in appetitive motivation[7, 8], as well as in aversive signalling [9, 10], among others, needs to be mentioned.

Lines 62-66: the mention to honey bees is not justified as the manuscript does not provide any data on honey bees. Moreover, the discussion of comparable honey bee findings is reduced to two lines in the Discussion (lines 405-407), which does not allow appreciating the level of similarity between the present findings and those obtained in honey bees.

Materials and Methods

Line 71: Referring to a work by Asada and Ono that readers cannot necessarily consult immediately is not helpful to understand the procedures used in this work. The authors should describe their procedures appropriately.

Line 88: same as above. Moreover, in Sasaki et al there is no proper illustration of the experimental setup used, and this is the critical point to understand the behavioral experiments on locomotor activity. I strongly recommend including a scheme of the behavioral ring-shaped chamber to understand what was quantified here.

Lines 189-192: Repetition of lines 181-183. Please check for coherence.

Line 304: should read “gynes” instead of workers; flupentixol was not assayed in workers, as indicated by the methods section and by the caption of Fig. 5.

Line 310: same as above; gynes instead of workers.

Line 313: “to the injected flupentixol”, and not dopamine.

Discussion

Expand to include more relevant information on findings related to the role of dopamine in social bees.

References

1. Schulz DJ, Pankiw T, Fondrk MK, Robinson GE, Page RE. Comparisons of juvenile hormone hemolymph and octopamine brain Titers in honey bees (Hymenoptera: Apidae) selected for high and low pollen hoarding. Annals of the Entomological Society of America. 2004;97(6):1313-9. doi: Doi 10.1603/0013-8746(2004)097[1313:Cojhha]2.0.Co;2. PubMed PMID: WOS:000225335000022.

2. Nouvian M, Mandal S, Jamme C, Claudianos C, d'Ettorre P, Reinhard J, et al. Cooperative defence operates by social modulation of biogenic amine levels in the honey bee brain. Proc Biol Sci. 2018;285(1871):20172653. Epub 2018/01/26. doi: 10.1098/rspb.2017.2653. PubMed PMID: 29367399; PubMed Central PMCID: PMCPMC5805953.

3. Iino S, Shiota Y, Nishimura M, Asada S, Ono M, Kubo T. Neural activity mapping of bumble bee (Bombus ignitus) brains during foraging flight using immediate early genes. Sci Rep. 2020;10(1):7887. doi: 10.1038/s41598-020-64701-1.

4. Yamane A, Kohno H, Ikeda T, Kaneko K, Ugajin A, Fujita T, et al. Gene expression and immunohistochemical analyses of mKast suggest its late pupal and adult-specific functions in the honeybee brain. PLoS One. 2017;12(5):e0176809. Epub 2017/05/05. doi: 10.1371/journal.pone.0176809. PubMed PMID: 28472083; PubMed Central PMCID: PMCPMC5417555.

5. Kaneko K, Suenami S, Kubo T. Gene expression profiles and neural activities of Kenyon cell subtypes in the honeybee brain: identification of novel 'middle-type' Kenyon cells. Zoological Lett. 2016;2:14. Epub 2016/08/02. doi: 10.1186/s40851-016-0051-6. PubMed PMID: 27478620; PubMed Central PMCID: PMCPMC4967334.

6. Geng H, Lafon G, Avarguès-Weber A, Buatois A, Massou I, Giurfa M. Visual learning in a virtual reality environment upregulates immediate early gene expression in the mushroom bodies of honey bees. Commun Biol. 2022;5(1):130. doi: 10.1038/s42003-022-03075-8.

7. Dong S, Gu G, Lin T, Wang Z, Li J, Tan K, et al. An inhibitory signal associated with danger reduces honeybee dopamine levels. Curr Biol. 2023;33. doi: 10.1016/j.cub.2023.03.072.

8. Huang J, Zhang Z, Feng W, Zhao Y, Aldanondo A, Sanchez MGdB, et al. Food wanting is mediated by transient activation of dopaminergic signaling in the honey bee brain. Science. 2022;376(6592):508-12. doi: doi:10.1126/science.abn9920.

9. Mustard JA, Vergoz V, Mesce KA, Klukas KA, Beggs KT, Geddes LH, et al. Dopamine Signaling in the Bee. In: Galizia CG, Eisenhardt D, Giurfa M, editors. Honeybee Neurobiology and Behavior: A Tribute to Randolf Menzel. Dordrecht: Springer Netherlands; 2012. p. 199-209.

10. Vergoz V, Roussel E, Sandoz JC, Giurfa M. Aversive learning in honeybees revealed by the olfactory conditioning of the sting extension reflex. PLoS One. 2007;2(3):e288. Epub 2007/03/21. doi: 10.1371/journal.pone.0000288. PubMed PMID: 17372627; PubMed Central PMCID: PMC1810431.

Reviewer #2: In order to investigate the roles of dopamine in regulating caste-specific behaviors and mating-related behaviors in bumble bees, the authors examined caste differences in behaviors, biogenic amine levels, and analyzed the effects of dopamine-related drugs on bumble bee behavior. The results showed that the locomotor and flight activities were significantly higher in 8-day-old gynes and light avoidance was significantly lower in 4–8-day-old gynes than in same-aged workers. Brain levels of dopamine and octopamine were significantly higher in 8-day-old gynes than in same-aged workers, but tyramine and serotonin levels did not differ between the castes. Dopamine significantly enhanced locomotor and flight activities in 7–9-day-old workers, whereas the dopamine receptor antagonist flupentixol inhibited flight activity and mating acceptance in same-aged gynes. These results suggested that dopamine played important roles in gyne-specific behavior in bumble bees. These results might be interesting for scientists in this field. The manuscript could be improved with minor revision.

1.The English of the whole paper should be polished by native English experts carefully.

2.The references should be corrected according to the requirement of this journal.

6. PLOS authors have the option to publish the peer review history of their article (what does this mean?). If published, this will include your full peer review and any attached files.

Reviewer #1: No

Reviewer #2: No

---

## [Author Response · Author response to Decision Letter 0]

15 Dec 2023

We have uploaded a file "Response to Reviewers". Please see the file.

---

## [Decision Letter · Decision Letter 1]

19 Jan 2024

PONE-D-23-31847R1Physiological specialization of the brain in bumble bee castes: Roles of dopamine in mating-related behaviors in female bumble beesPLOS ONE

Dear Dr. Sasaki,

Thank you for submitting your manuscript to PLOS ONE. After careful consideration, we feel that it has merit but does not fully meet PLOS ONE’s publication criteria as it currently stands. Therefore, we invite you to submit a revised version of the manuscript that addresses the points raised during the review process.

The revisions you have made considering feedback from both reviewers are strong, and I have had the manuscript re reviewed. There remains a few minor points that one reviewer would like clarified, and if you can enable these final few points, then there is good support for publication. 

We look forward to receiving your revised manuscript.

Kind regards,

Adrian G Dyer, Ph.D.

Academic Editor

PLOS ONE

Journal Requirements:

Reviewers' comments:

Reviewer's Responses to Questions

**Comments to the Author**

1. If the authors have adequately addressed your comments raised in a previous round of review and you feel that this manuscript is now acceptable for publication, you may indicate that here to bypass the “Comments to the Author” section, enter your conflict of interest statement in the “Confidential to Editor” section, and submit your "Accept" recommendation.

Reviewer #1: All comments have been addressed

2. Is the manuscript technically sound, and do the data support the conclusions?

Reviewer #1: Yes

3. Has the statistical analysis been performed appropriately and rigorously? 

Reviewer #1: Yes

4. Have the authors made all data underlying the findings in their manuscript fully available?

Reviewer #1: Yes

5. Is the manuscript presented in an intelligible fashion and written in standard English?

Reviewer #1: No

6. Review Comments to the Author

Reviewer #1: I thank the authors for having addressed all my comments. Although I may not necessarily subscribe some arguments justifying the whole brain approach for molecular analyses related to behavior, I understand the arguments provided by the authors and I consider that they are valid. I will not, therefore, further insist on this point, as the results presented in the manuscript are valuable and worth being published.

I have only some very minor remarks that can be easily corrected by the authors in order to better convey their message and ideas.

I am listing these corrections below:

Line 77: “…was dived IN TWO sections…”

Line 107: “… can walk freely within the ANNULAR circuit (gray area, WHICH WAS COVERED BY A transparent cover sheet. THE SHEET was separated….”

In the description of the experimental setup as well in the new figure itself; the height of the setup containing the bumble bee is missing. This is important to appreciate how were the flight attempts when the red cover was suppressed.

Line 215: “… behavioral experiments DESCRIBED above”.

Line 410: “…detected BY several sense organs…”.

Line 412: “ … unified analytical levels among experiments”. I don’t understand what is meant here. Please use another wording to make the idea clearer.

Line 416-417: “…a suitable first step THAT SHOULD BE FOLLOWED BY A BRAIN-REGION BASED APPROACH.”

Line 450: “Dopamine PLAYS OTHER ROLES IN THE BEHAVIOR OF SOCIAL INSECTS”. IN HONEY BEES, IT MEDIATES A DECREASE OF SUCROSE RESPONSIVENESS [11; BUT SEE HUANG ET AL, SCIENCE 2022, FIG. 3], AFFECTS MEMORY RETRIEVAL IN AN APPETITIVE CONTEXT [12, 13] , REGULATES APPETITIVE MOTIVATION [13,14] AND MEDIATES AVERSIVE REINFORCEMENT SIGNALING IN AVERSIVE LEARNING [15,16]. IN ANTS, DOPAMINE IS RELEVANT FOR THE DEFENSE OF THE TERRITORY OR COLONY [35].”

7. PLOS authors have the option to publish the peer review history of their article (what does this mean?). If published, this will include your full peer review and any attached files.

Reviewer #1: No

---

## [Author Response · Author response to Decision Letter 1]

22 Jan 2024

We have uploaded a file "Response to Reviewers". Please see the file.

---

## [Editor Report · Decision Letter 2]

30 Jan 2024

Physiological specialization of the brain in bumble bee castes: Roles of dopamine in mating-related behaviors in female bumble bees

PONE-D-23-31847R2

Dear Dr. Sasaki,

We’re pleased to inform you that your manuscript has been judged scientifically suitable for publication and will be formally accepted for publication once it meets all outstanding technical requirements.

Kind regards,

Adrian G Dyer, Ph.D.

Academic Editor

PLOS ONE
---

## [Editor Report · Acceptance letter]

3 Mar 2024

PONE-D-23-31847R2 

PLOS ONE

Dear Dr. Sasaki, 

I'm pleased to inform you that your manuscript has been deemed suitable for publication in PLOS ONE. Congratulations! Your manuscript is now being handed over to our production team.

Kind regards, 

on behalf of

Dr. Adrian G Dyer 

Academic Editor

PLOS ONE